# An Attribute Reduction Method Using Neighborhood Entropy Measures in Neighborhood Rough Sets

**DOI:** 10.3390/e21020155

**Published:** 2019-02-07

**Authors:** Lin Sun, Xiaoyu Zhang, Jiucheng Xu, Shiguang Zhang

**Affiliations:** 1College of Computer and Information Engineering, Henan Normal University, Xinxiang 453007, China; 2Engineering Technology Research Center for Computing Intelligence and Data Mining, Henan 453007, China

**Keywords:** rough sets, neighborhood rough sets, attribute reduction, neighborhood entropy, classification

## Abstract

Attribute reduction as an important preprocessing step for data mining, and has become a hot research topic in rough set theory. Neighborhood rough set theory can overcome the shortcoming that classical rough set theory may lose some useful information in the process of discretization for continuous-valued data sets. In this paper, to improve the classification performance of complex data, a novel attribute reduction method using neighborhood entropy measures, combining algebra view with information view, in neighborhood rough sets is proposed, which has the ability of dealing with continuous data whilst maintaining the classification information of original attributes. First, to efficiently analyze the uncertainty of knowledge in neighborhood rough sets, by combining neighborhood approximate precision with neighborhood entropy, a new average neighborhood entropy, based on the strong complementarity between the algebra definition of attribute significance and the definition of information view, is presented. Then, a concept of decision neighborhood entropy is investigated for handling the uncertainty and noisiness of neighborhood decision systems, which integrates the credibility degree with the coverage degree of neighborhood decision systems to fully reflect the decision ability of attributes. Moreover, some of their properties are derived and the relationships among these measures are established, which helps to understand the essence of knowledge content and the uncertainty of neighborhood decision systems. Finally, a heuristic attribute reduction algorithm is proposed to improve the classification performance of complex data sets. The experimental results under an instance and several public data sets demonstrate that the proposed method is very effective for selecting the most relevant attributes with great classification performance.

## 1. Introduction

Attribute reduction in rough set theory has been recognized as an important feature selection method, aimed to select the most representative attribute subset with a high resolution by eliminating redundant and unimportant attributes [1]. The attribute reduction methods can be widely implemented in the fields of data classification, data mining, machine learning, and pattern recognition [2,3,4,5,6]. Due to the development of the internet, the scale of data becomes bigger and bigger. Even thousands of attributes may be acquired in some real-world databases. In order to shorten the processing time and obtain better generalization, the attribute reduction problem attracts more and more attention in recent years [5,7,8].

In the classical rough set theory, there are two forms of definition for attribute reduction. The one is the algebra definition based on approximate precision, which determines whether certain conditional attributes can be removed according to the variation of approximate precision and considers the effect of attributes on the deterministic subsets in the field. The other is the definition of information view based on information entropy, which determines whether certain conditional attributes can be removed according to the changes of conditional entropy and considers the effect of attributes on the indeterminate subsets in the field [9]. Many attribute reduction algorithms are based on the algebra definition so far. Mi et al. [10] introduced the concepts of a lower distribution reduct and an upper distribution reduct based on the variable precision rough sets, and obtained an approach for knowledge reduction in variable precision rough sets. Syau et al. [11] provided the characterizations of lower and upper approximations for the connection between the concepts of variable precision generalized rough set model and neighborhood systems by introducing minimal neighborhood systems. What is more, as a measure to evaluate the uncertainty of discrete sample spaces, information entropy is a significant tool for characterizing the distinguishment information of attributes subsets [12]. Information entropy based on neighborhood systems has been established, and the extension of information entropy and its variants are adapted for attribute reduction. Gao et al. [13] developed a heuristic attribute reduction algorithm based on the maximum decision entropy in the decision-theoretic rough set model. Dai et al. [14] proposed a framework for attribute reduction in interval-valued data from the information view. It is known that there is a strong complementarity between the algebra view and the information view of attribute importance, and the two views can be combined to produce a more comprehensive measurement mechanism [15]. Wang [15] summarized the reduct in rough sets from algebra view and information view. This inspires the authors to investigate new attribute reduction methods from algebra view and information view in this paper.

The classical rough set theory is established on the equivalence approximate space and only compatible for discrete data set, and it could be useless for continuous numerical data [13,15,16]. In general, it needs to discretize when processing continuous numerical data, which will lead to the loss of information (including the neighborhood structure information and order structure information in real spaces) [17,18]. To overcome this drawback, many extensions of classical rough set theory have been presented [19,20,21,22,23,24,25,26], such as fuzzy rough set [21,22], tolerance approximate models [23], similarity rough approximate model [24], covering approximation model [25], and neighborhood granular model [26]. Among all the extensions, Hu et al. [18] developed a neighborhood rough set model to process both numerical and categorical data sets via neighborhood relation. Then, the neighborhood rough set model can process both numerical and discrete data sets via neighborhood parameters. Wang et al. [27] presented a local neighborhood rough set combining the neighborhood rough set and local rough set, to be applied to rough data analysis in big data. Fan et al. [28] designed an attribute reduction algorithm based on the max-decision neighborhood rough set model. Chen et al. [29] investigated approaches to attribute reduction in parallel using dominance-based neighborhood rough sets. Therefore, this paper studies an attribute reduction algorithm based on neighborhood rough sets by making full use of the advantages for rough sets.

It is noted that the reduction calculation of decision neighborhood systems is a key problem in neighborhood rough set theory. In addition, the reducts of an information system need to be achieved to further extract rule-like knowledge from information systems [29]. In practical application of decision-making, both the certainty factor and the object coverage factor of rule are two important standards of evaluating the decision ability of decision systems [30,31]. However, some of these existing knowledge reduction methods cannot reflect the change of decision ability for classification objectively. It is known that the credibility degree and the coverage degree can efficiently reflect the classification ability of conditional attributes with respect to the decision attribute [30]. The conditional attributes with higher credibility and coverage degrees are more important with respect to the decision attribute. Therefore, it is necessary to investigate a new uncertainty measure and an effective heuristic search algorithm. Sun et al. [32] proposed a concept of decision degree based on the notions of the certainty factor and the coverage factor of rule in rough sets, which includes the degree of sufficiency of a proposition and the degree of its necessity. Until now, the works are not considered in neighborhood rough sets. This inspires the authors to investigate a new measure to effectively illustrate the classification ability and the decision ability of neighborhood decision systems. Based on this idea, the concepts of coverage and credibility degrees are introduced into neighborhood decision systems as measures to reflect the classification ability of conditional attributes, with respect to decision attributes in this paper. In order to fully reflect the decision ability of attributes, the credibility degree and the coverage degree based on neighborhood rough sets are integrated into neighborhood entropy measures. Then, a heuristic attribute reduction method based on decision neighborhood entropy is presented to address the uncertainty and noisiness of complex data sets in neighborhood rough sets. 

The remainder of this paper is organized as follows: Section 2 reviews some basic concepts of rough sets, information entropy measures, and neighborhood rough sets. In Section 3, some uncertainty measures based on neighborhood entropy in neighborhood decision systems are investigated, their properties are derived, and the relationships among these measures are established. An attribute reduction algorithm based on decision neighborhood entropy with complexity analysis is designed. Section 4 analyzes the classification experiments conducted on several public data sets. Finally, Section 5 summarizes the study.

## 2. Previous Knowledge

In this section, we briefly review several basic concepts of rough sets, information entropy measures and neighborhood rough sets in [12,13,18,33,34,35].

### 2.1. Rough Sets

Given a decision system *DS* = (*U*, *C*, *D*, *V*, *f*), usually written more simply as *DS* = (*U*, *C*, *D*), where *U* = {*x*_1_, *x*_2_, ⋯, *x_n_*} is a sample set named universe, *C* = {*a*_1_, *a*_2_, ⋯, *a_m_*} is a conditional attribute set that describes the samples, *D* is a set of classification attributes, *f*: *U* × {*C*∪*D*}→*V* is an information function which associates a unique value of each attribute with every object belonging to *U*, and *f*(*a*, *x*) represents the value of *x* ∈ *U* on attribute *a* ∈ *C*∪*D*. For any *B*⊆*C*, two samples *x*, *y* ∈ *U*, the equivalence relation is described as
*IND*(*B*) = {(*x*, *y*)|∀*a* ∈ *B*, *f*(*a*, *x*) = *f*(*a*, *y*)}.(1)

Then, *U*/*IND*(*B*) is called a partition that is composed of the equivalence classes, and for any sample *x* ∈ *U*, [*x*]*_B_* = {*y*| *y* ∈ *U*, (*x*, *y*) ∈ *IND*(*B*)} is an equivalence class of *x*.

In a decision system *DS* = (*U*, *C*, *D*) with *B*⊆*C* and *X*⊆*U*, the lower approximation set and the upper approximation set of *X* with respect to *B* can be expressed, respectively, as
(2)$$\underline{B}(X)=\{x|{\left[x\right]}_{B}\subseteq X,x\in U\},$$
(3)$$\bar{B}(X)=\{x|{\left[x\right]}_{B}\cap X\neq \emptyset ,x\in U\}.$$

### 2.2. Information Entropy Measures

Given a decision system *DS* = (*U*, *C*, *D*) with *B*⊆*C*, and *U*/*B* = {*X*_1_, *X*_2_, ⋯, *X**_N_*}, then the information entropy of *B* is described as
(4)$$H(B)=-\sum_{i=1}^{{}_{N}}p(X_{i})\log p(X_{i}),$$
where $p\left(X_{i}\right)=\frac{\left|X_{i}\right|}{\;\left|U\right|}$ is the probability of *X_i_*⊆*U*/*B*, and |*X_i_*| denotes the cardinality of the equivalence class *X_i_*.

Given a decision system *DS* = (*U*, *C*, *D*) with *B*_1_, *B*_2_⊆*C*, *U*/*B*_1_ = {*X*_1_, *X*_2_, ⋯, *X**_N_*}, and *U*/*B*_2_ = {*Y*_1_, *Y*_2_, ⋯, *Y**_M_*}, then the joint entropy of *B*_1_ and *B*_2_ is denoted as
(5)$$H(B_{1}\cup B_{2})=-\sum_{i=1}^{N}\sum_{j=1}^{M}p(X_{i}\cap Y_{j})\log p(X_{i}\cap Y_{j}),$$
where $p\left(X_{i}\cap Y_{j}\right)=\frac{\left|X_{i}\cap Y_{j}\right|}{\;\left|U\right|}$, *i* = 1, 2,···, *N*, and *j* = 1, 2,···, *M*.

Given a decision system *DS* = (*U*, *C*, *D*) with *B*_1_, *B*_2_⊆*C*, *U*/*B*_1_ = {*X*_1_, *X*_2_, ⋯, *X**_N_*}, and *U*/*B*_2_ = {*Y*_1_, *Y*_2_, ⋯, *Y**_M_*}, then the conditional information entropy of *B*_2_ with respect to *B*_1_ is defined as
(6)$$H(B_{2}|B_{1})=-\sum_{i=1}^{N}p(X_{i})\sum_{j=1}^{M}p(Y_{j}|X_{i})\log p(Y_{j}|X_{i}),$$
where $p\left(Y_{j}|X_{i}\right)=\frac{\left|Y_{j}\cap X_{i}\right|}{\;\left|X_{i}\right|}$, *i* = 1, 2, ⋯, *N*, and *j* = 1, 2, ⋯, *M*.

Given a decision system *DS* = (*U*, *C*, *D*) with *B*_1_, *B*_2_⊆*C*, the mutual information between *B*_1_ and *B*_2_ is defined as
*I*(*B*_1_; *B*_2_) = *H*(*B*_2_) − *H*(*B*_2_|*B*_1_).(7)

Given a decision system *DS* = (*U*, *C*, *D*) with *B*_1_, *B*_2_⊆*C*, the mutual information has the following properties: (1)*I*(*B*_1_; *B*_2_) ≥ 0,(2)*I*(*B*_1_; *B*_2_) = *I*(*B*_2_; *B*_1_),(3)*I*(*B*_1_; *B*_2_) = *H*(*B*_1_) + *H*(*B*_2_) − *H*(*B*_1_∪*B*_2_),(4)*I*(*B*_1_; *B*_2_) = *H*(*B*_1_) − *H*(*B*_1_|*B*_2_) = *H*(*B*_2_) − *H*(*B*_2_|*B*_1_).

Given a decision system *DS* = (*U*, *C*, *D*) with *B*⊆*C*, if *I*(*B*; *D*) = *I*(*C*; *D*) and *I*(*B* − {*a*}; *D*) < *I*(*B*; *D*) for any *a*∈*B*, then *B* is a reduct of *C* with respect to *D*.

### 2.3. Neighborhood Rough Sets

Given a neighborhood decision system *NDS* = (*U*, *C*, *D*, *V*, *f*, ∆, *δ*), usually written more simply as *NDS* = (*U*, *C*, *D*, *δ*), where *U* = {*x*_1_, *x*_2_, ⋯, *x_n_*} is a sample set named universe, *C* = {*a*_1_, *a*_2_, ⋯, *a_m_*} is a conditional attribute set that describes the samples, *D* = {*d*} is a decision attribute set that contains only one decision attribute, *V* = $\cup {}_{a\in \{C\cup D\}}$*_D_*_}_*V_a_*, *V_a_* is a value set of attribute *a*, *f*: *U* × {*C*∪*D*}→*V* is a map function, ∆→[0, ∞) is a distance function, and *δ* is a neighborhood parameter with 0 ≤ *δ* ≤ 1. 

For any samples *x*, *y*, *z*∈*U* on a subset *B*, the distance function ∆*_B_*(*x*, y) satisfies the following three conditions: ∆*_B_*(*x*, *y*) ≥ 0, ∆*_B_*(*x*, *y*) = ∆*_B_*(*y*, *x*), and ∆*_B_*(*x*, *y*) + ∆*_B_*(*y*, *z*) ≥ ∆*_B_*(*x*, *z*).

It is well known that there are three classical metrics including Manhattan, Euclidean, and Chebychev distance functions, where the Euclidean distance function effectively reflects the basic information of the unknown data [33]. Given a neighborhood decision system *NDS* = (*U*, *C*, *D*, *δ*) with *B*⊆*C*, for any *x*, *y*∈*U*, the Euclidean distance function between *x* and *y* is expressed as
(8)$$\Delta_{B}(x,y)=\sqrt{\sum_{k=1}^{\left|B\right|}{\left|f(a_{k},x)-f(a_{k},y)\right|}^{2}}.$$

Given a neighborhood decision system *NDS* = (*U*, *C*, *D*, *δ*) with *B*⊆*C*, the similarity relation resulting by *B* is defined as
(9)$$NR_{\delta }(B)=\{(x,y)\in U\times U|\Delta_{B}(x,y)\leq \delta \}.$$

Given a neighborhood decision system *NDS* = (*U*, *C*, *D*, *δ*) with *B*⊆*C*, for any *x* ∈ *U*, the neighborhood class of *x* with respect to *B* is described as
(10)$$n_{B}^{\delta }(x)=\{y|x,y\in U,\Delta_{B}(x,y)\leq \delta \}.$$

Given a neighborhood decision system *NDS* = (*U*, *C*, *D*, *δ*) with *B*⊆*C* and *X*⊆*U*, the neighborhood lower approximation set and the neighborhood upper approximation set of *X* with respect to *B* are described, respectively, as
(11)$${\underline{B}}_{\delta }(X)=\{x_{i}|n_{B}^{\delta }(x_{i})\subseteq X,\;x_{i}\in U,\;i=1,\;2,\;\cdots ,\;n\},$$
(12)$${\bar{B}}_{\delta }(X)=\{x_{i}|n_{B}^{\delta }(x_{i})\cap X\neq \emptyset ,x_{i}\in U,\;i=1,\;2,\;\cdots ,\;n\}.$$

Given a neighborhood decision system *NDS* = (*U*, *C*, *D*, *δ*) with *B*⊆*C* and *X*⊆*U*, ${\underline{B}}_{\delta }(X)$ is the neighborhood lower approximation set of *X* with respect to *B*, and ${\bar{B}}_{\delta }(X)$ is the neighborhood upper approximation set of *X* with respect to *B*, then the approximate precision of *X* with respect to *B* is described as
(13)$$p_{B}(X)=\frac{\left|{\underline{B}}_{\delta }(X)\right|}{\left|{\bar{B}}_{\delta }(X)\right|}.$$

Given a neighborhood decision system *NDS* = (*U*, *C*, *D*, *δ*) with *B*⊆*C*, *U*/*D* = {*X*_1_, *X*_2_, ⋯, *X**_N_*}, then the neighborhood lower approximation set and the neighborhood upper approximation set of *D* with respect to *B* are described respectively as
(14)$${\underline{B}}_{\delta }(D)=\overset{N}{\underset{i=1}{\cup }}{\underline{B}}_{\delta }(X_{i}),$$
(15)$${\bar{B}}_{\delta }(D)=\overset{N}{\underset{i=1}{\cup }}{\bar{B}}_{\delta }(X_{i}),$$
where ${\underline{B}}_{\delta }(X_{i})$ respects the neighborhood lower approximation set of *X_i_* with respect to *B*, ${\bar{B}}_{\delta }(X_{i})$ respects the neighborhood upper approximation set of *X**_i_* with respect to *B*, and *i* = 1, 2, ⋯, *N*.

Given a neighborhood decision system *NDS* = (*U*, *C*, *D*, *δ*) with *B*⊆*C*, then the neighborhood approximate precision of *D* with respect to *B* is described as
(16)$$p_{B}(D)=\frac{\left|{\underline{B}}_{\delta }(D)\right|}{\left|{\bar{B}}_{\delta }(D)\right|}.$$

The neighborhood approximate precision can be used to reflect the complete degree of the knowledge of a set, but this precision measure does not take into account the size of the particles that are included in the lower approximation set completely. Therefore, it is not sufficient to only consider attribute reduction from the algebra view.

## 3. Attribute Reduction Method Using Neighborhood Entropy Measures in Neighborhood Decision Systems

Attribute reduction is a core part of the rough set theory [13]. In the classical rough set theory, there are two forms of definition for attribute reduction: One is the algebra definition based on set theory; the other is the definition of information view based on information entropy. There is a strong complementarity between the algebra definition of attribute significance and the definition of information view. The former considers the influence of attributes on the defined subset in the domain of theory, while the latter considers the influence of attributes on the uncertain subset in the domain of theory. Therefore, the two views can be combined to produce a more comprehensive measurement mechanism. In rough sets, the equivalence classes-based information entropy does not work for numerical data. Then, for continuous data sets, a discretization should be performed before further processing. However, the discretization may result in information loss, and it is difficult to employ mutual information in attribute evaluation due to the difficulty in estimating the probability density of attributes [36]. To address this issue, the concept of neighborhood can be combined with information theory to extend Shannon entropy, and then some correlative concepts of neighborhood entropy are defined to measure the uncertainty of numerical data. Then, some concepts of neighborhood entropy-based uncertainty measures are presented to measure the uncertainty of knowledge in neighborhood decision systems, some important properties and relationships of these measures are deduced respectively as well, and a heuristic attribute reduction algorithm is investigated to improve the classification performance of complex data sets.

### 3.1. Neighborhood Entropy-Based Uncertainty Measures

Given a neighborhood decision system *NDS* = (*U*, *C*, *D*, *δ*) with *B*⊆*C*, $n_{B}^{\delta }(x_{i})$ is a neighborhood class of *x_i_* ∈ *U*, then Hu et al. [37] described the neighborhood entropy of *x_i_* as
(17)$$H_{\delta }^{x_{i}}(B)=-\log\frac{\left|n_{B}^{\delta }(x_{i})\right|}{\left|U\right|}.$$

Given a neighborhood decision system *NDS* = (*U*, *C*, *D*, *δ*) with *B*⊆*C*, Hu et al. [37] and Chen et al. [33] computed the average neighborhood entropy of the sample set as
(18)$$H_{\delta }(B)=-\frac{1}{\left|U\right|}\sum_{i=1}^{\left|U\right|}\log\frac{\left|n_{B}^{\delta }(x_{i})\right|}{\left|U\right|}.$$

The concept of neighborhood entropy is defined based on the information entropy theory, which granulates the space of the domain by neighborhood relation and is used to measure the uncertainty and classification ability of the numerical knowledge classification system [38]. In this paper, the neighborhood approximate precision is combined with the neighborhood entropy to reflect the uncertainty of knowledge, and then a new average neighborhood entropy is defined as follows.

**Definition** **1.**
*Given a neighborhood decision system NDS = (U, C, D, δ) with B⊆C, a new average neighborhood entropy of the sample set is defined as*
(19)$$H_{\delta }(B)=-\frac{p_{B}(D)}{\left|U\right|}\sum_{i=1}^{\left|U\right|}\log\frac{\left|n_{B}^{\delta }(x_{i})\right|}{\left|U\right|}.$$


From Definition 1, the average neighborhood entropy combines the neighborhood precision with the average neighborhood entropy, and it can make full use of the advantages of algebra and information view, and overcomes the drawbacks of traditional precision measurement.

**Property** **1.**
*Given a neighborhood decision system NDS = (U, C, D, δ) with x_i_∈U, then 0 ≤ H_δ_ (C) ≤ log|U|.*


**Proof.** It follows from Equation (10) that $n_{C}^{\delta }(x_{i})$⊆*U* for any *x_i_*∈*U*, then one has that $\frac{1}{\left|U\right|}\leq \frac{\left|n_{C}^{\delta }(x_{i})\right|}{\left|U\right|}\leq 1$. From Equation (16), it is obtained that $0\leq p_{B}(D)\leq 1$, and then $0\leq \frac{p_{B}(D)}{\left|U\right|}\leq \frac{1}{\left|U\right|}$. Thus, it is obvious from Definition 1 that 0 ≤ *H_δ_* (*C*) ≤ log|*U*|. □

**Proposition** **1.***Given a neighborhood decision system NDS = (U, C, D, δ) with x_i_*∈*U, if B_1_*⊆*B_2_*⊆*C, then*$H_{\delta }(B_{2})\geq H_{\delta }(B_{1})$.

**Proof.** Suppose that *B*_1_⊆*B*_2_⊆*C*, and similar to the proof of Proposition 1 in [33], one has that $n_{B_{1}}^{\delta }(x_{i})⊇n_{B_{2}}^{\delta }(x_{i})$. Then, $\left|n_{B_{1}}^{\delta }(x_{i})\right|\geq \left|n_{B_{2}}^{\delta }(x_{i})\right|$ holds. It follows from Equations (11) and (12) that ${\underline{B_{1}}}_{\delta }(X)\subseteq {\underline{B_{2}}}_{\delta }(X)$ and ${\bar{B_{1}}}_{\delta }(X)⊇{\bar{B_{2}}}_{\delta }(X)$. By Equation (16), one has that $p_{B_{1}}(D)\leq p_{B_{2}}(D)$. Hence, it can be obtained from Equation (19) that $H_{\delta }(B_{2})\geq H_{\delta }(B_{1})$. □

**Definition** **2.**
*Given a neighborhood decision system NDS = (U, C, D, δ) with B*
⊆
*C,*
$n_{B}^{\delta }(x_{i})$
*is a neighborhood class of x_i_*
∈
*U generated by*
*NR*
*_δ_*
*(B)*
*, and [x_i_]_D_ is an equivalence class of x_i_*
∈
*U generated by IND(D), then a decision neighborhood entropy of B and D is defined as*
(20)$$H_{\delta }(D,B)=-\frac{p_{B}(D)}{\left|U\right|}\sum_{i=1}^{\left|U\right|}\log(\frac{{\left|n_{B}^{\delta }(x_{i})\cap {\left[x_{i}\right]}_{D}\right|}^{2}}{\left|U\right|\left|{\left[x_{i}\right]}_{D}\right|}).$$


In a decision system *DS* = (*U*, *C*, *D*) with any *x_i_*∈*U*, Pawlak et al. [34] and Wang et al. [30] express a decision rule as $d_{x_{i}}:\;des({\left[x_{i}\right]}_{C})\Rightarrow des({\left[x_{i}\right]}_{D})(\beta_{x_{i}},\lambda_{x_{i}})$, where $des({\left[x_{i}\right]}_{C})$ and $des({\left[x_{i}\right]}_{D})$ are the descriptions of *x_i_* under the equivalence relations *IND*(*C*) and *IND*(*D*), respectively. $\beta_{x_{i}}=\frac{\left|{\left[x_{i}\right]}_{C}\cap {\left[x_{i}\right]}_{D}\right|}{\left|{\left[x_{i}\right]}_{C}\right|}$ is called the credibility degree of decision rule $d_{x_{i}}$, and $\lambda_{x_{i}}=\frac{\left|{\left[x_{i}\right]}_{C}\cap {\left[x_{i}\right]}_{D}\right|}{\left|U\right|}$ is called the coverage degree of decision rule $d_{x_{i}}$. Wang et al. [30] declared that the credibility degree and the coverage degree can reflect the classification ability of conditional attributes with respect to the decision attribute, and the conditional attributes with higher credibility and coverage degrees are more important with respect to the decision attribute. Furthermore, Tsumoto [31] emphasized that the credibility degree indicates the adequacy of the proposition, and the coverage degree describes the necessity of the proposition. Then, in order to fully reflect the decision ability and the classification ability of neighborhood decision systems, this paper investigates some neighborhood entropy-based uncertainty measures by combining the credibility degree with the coverage degree in neighborhood rough sets.

**Property** **2.***Given a neighborhood decision system NDS = (U, C, D, δ),**if*$\beta_{x_{i}}=\frac{\left|n_{B}^{\delta }(x_{i})\cap {\left[x_{i}\right]}_{D}\right|}{\left|{\left[x_{i}\right]}_{D}\right|}$*and*$\lambda_{x_{i}}=\frac{\left|n_{B}^{\delta }(x_{i})\cap {\left[x_{i}\right]}_{D}\right|}{\left|U\right|}$*for**any**x_i_*∈*U**,**then one has that*$H_{\delta }(D,B)=-\frac{p_{B}(D)}{\left|U\right|}\sum_{i=1}^{\left|U\right|}\log(\beta_{x_{i}}\lambda_{x_{i}})$.

**Proof.** It follows immediately from Definition 2 that
$$\begin{array}{ll} H_{\delta }(D,B) & =-\frac{p_{B}(D)}{\left|U\right|}\sum_{i=1}^{\left|U\right|}\log(\frac{{\left|n_{B}^{\delta }(x_{i})\cap {\left[x_{i}\right]}_{D}\right|}^{2}}{\left|U\right|\left|{\left[x_{i}\right]}_{D}\right|})\\ & =-\frac{p_{B}(D)}{\left|U\right|}\sum_{i=1}^{\left|U\right|}\log(\frac{\left|n_{B}^{\delta }(x_{i})\cap {\left[x_{i}\right]}_{D}\right|}{\left|U\right|}\cdot \frac{\left|n_{B}^{\delta }(x_{i})\cap {\left[x_{i}\right]}_{D}\right|}{\left|{\left[x_{i}\right]}_{D}\right|})\\ & =-\frac{p_{B}(D)}{\left|U\right|}\sum_{i=1}^{\left|U\right|}\log(\beta_{x_{i}}\lambda_{x_{i}}).\;□ \end{array}$$

Property 2 shows that the decision neighborhood entropy of *B* and *D* combines the credibility degree and the coverage degree in the neighborhood decision system, which can fully reflect the decision ability of the neighborhood decision system.

**Proposition** **2.***Given a neighborhood decision system NDS = (U, C, D, δ) with B_1_*⊆*B_2_*⊆*C, then*$H_{\delta }(D,B_{1})\;\leq H_{\delta }\left(D,B_{2}\right)$*, where the equal sign holds if and only if*$n_{B_{1}}^{\delta }(x_{i})=n_{B_{2}}^{\delta }(x_{i})$*for**any**x_i_*∈U.

**Proof.** Suppose that *B*_1_⊆*B*_2_⊆*C*, it follows from Proposition 1 that $n_{B_{1}}^{\delta }(x_{i})⊇n_{B_{2}}^{\delta }(x_{i})$. Then, it is obvious that $U⊇n_{B_{1}}^{\delta }(x_{i})\cap {\left[x_{i}\right]}_{D}⊇n_{B_{2}}^{\delta }(x_{i})\cap {\left[x_{i}\right]}_{D}⊇\{x_{i}\}$. It is easily obtained that $\left|U\right|\geq \left|n_{B_{1}}^{\delta }(x_{i})\cap {\left[x_{i}\right]}_{D}\right|\geq$
$\left|n_{B_{2}}^{\delta }(x_{i})\cap {\left[x_{i}\right]}_{D}\right|\geq \left|\left\{x_{i}\right\}\right|=1$. Thus, one has that $\frac{{\left|U\right|}^{2}}{\left|U\right|\left|{\left[x_{i}\right]}_{D}\right|}\geq \frac{{\left|n_{B_{1}}^{\delta }(x_{i})\cap {\left[x_{i}\right]}_{D}\right|}^{2}}{\left|U\right|\left|{\left[x_{i}\right]}_{D}\right|}\geq \frac{{\left|n_{B_{2}}^{\delta }(x_{i})\cap {\left[x_{i}\right]}_{D}\right|}^{2}}{\left|U\right|\left|{\left[x_{i}\right]}_{D}\right|}\geq \frac{1}{\left|U\right|\left|{\left[x_{i}\right]}_{D}\right|}$. So, $\log(\frac{\left|U\right|}{\left|{\left[x_{i}\right]}_{D}\right|})\geq \log(\frac{{\left|n_{B_{1}}^{\delta }(x_{i})\cap {\left[x_{i}\right]}_{D}\right|}^{2}}{\left|U\right|\left|{\left[x_{i}\right]}_{D}\right|})\geq \log(\frac{{\left|n_{B_{2}}^{\delta }(x_{i})\cap {\left[x_{i}\right]}_{D}\right|}^{2}}{\left|U\right|\left|{\left[x_{i}\right]}_{D}\right|})\geq \log(\frac{1}{\left|U\right|\left|{\left[x_{i}\right]}_{D}\right|})$ obviously holds. In addition, from Equations (11) and (12), it follows that ${\underline{B_{1}}}_{\delta }(X)\subseteq {\underline{B_{2}}}_{\delta }(X)$ and ${\bar{B_{1}}}_{\delta }(X)⊇{\bar{B_{2}}}_{\delta }(X)$. According to Equation (16), one has that $p_{B_{1}}(D)\leq p_{B_{2}}(D)$. Hence, it can be obtained from Definition 2 that $H_{\delta }\left(D,B_{1}\right)\leq H_{\delta }\left(D,B_{2}\right)$. When $n_{B_{1}}^{\delta }(x_{i})=n_{B_{2}}^{\delta }(x_{i})$ for any *x_i_*∈*U*, it is obvious that $\frac{{\left|n_{B_{1}}^{\delta }(x_{i})\cap {\left[x_{i}\right]}_{D}\right|}^{2}}{\left|U\right|\left|{\left[x_{i}\right]}_{D}\right|}=\frac{{\left|n_{B_{2}}^{\delta }(x_{i})\cap {\left[x_{i}\right]}_{D}\right|}^{2}}{\left|U\right|\left|{\left[x_{i}\right]}_{D}\right|}$, and one has $p_{B_{1}}(D)=p_{B_{2}}(D)$. From Definition 2, it follows that $H_{\delta }\left(D,B_{1}\right)\;=\;H_{\delta }\left(D,B_{2}\right)$. Therefore, $H_{\delta }\left(D,B_{1}\right)\leq$
$H_{\delta }\left(D,B_{2}\right)$ holds. □

The monotonicity is one of the most important properties for an effective uncertainty measure of attribute reduction. According to Proposition 2, it is quite obvious that the decision neighborhood entropy is monotonic, decreasing when adding the condition attributes, which validates the monotonicity of the proposed uncertainty measure.

**Proposition** **3.**
*Given a neighborhood decision system NDS = (U, C, D, δ) with B⊆C, then H_δ_ (D, B) ≥ H_δ_ (B).*


**Proof.** It follows immediately from Definitions 1 and 2 that:$$\begin{array}{l} H_{\delta }(D,B)-H_{\delta }\left(B\right)\\ =-\frac{p_{B}(D)}{\left|U\right|}\sum_{i=1}^{\left|U\right|}\log(\frac{{\left|n_{B}^{\delta }(x_{i})\cap {\left[x_{i}\right]}_{D}\right|}^{2}}{\left|U\right|\left|{\left[x_{i}\right]}_{D}\right|})+\frac{p_{B}(D)}{\left|U\right|}\sum_{i=1}^{\left|U\right|}\log\frac{\left|n_{B}^{\delta }(x_{i})\right|}{\left|U\right|}\\ =-\frac{p_{B}(D)}{\left|U\right|}\sum_{i=1}^{\left|U\right|}\log(\frac{{\left|n_{B}^{\delta }(x_{i})\cap {\left[x_{i}\right]}_{D}\right|}^{2}}{\left|U\right|\left|{\left[x_{i}\right]}_{D}\right|}\cdot \frac{\left|U\right|}{\left|n_{B}^{\delta }(x_{i})\right|})\\ =-\frac{p_{B}(D)}{\left|U\right|}\sum_{i=1}^{\left|U\right|}\log(\frac{{\left|n_{B}^{\delta }(x_{i})\cap {\left[x_{i}\right]}_{D}\right|}^{2}}{\left|{\left[x_{i}\right]}_{D}\right|\left|n_{B}^{\delta }(x_{i})\right|})\\ =-\frac{p_{B}(D)}{\left|U\right|}\sum_{i=1}^{\left|U\right|}\log(\frac{\left|n_{B}^{\delta }(x_{i})\cap {\left[x_{i}\right]}_{D}\right|}{\left|{\left[x_{i}\right]}_{D}\right|}\cdot \frac{\left|n_{B}^{\delta }(x_{i})\cap {\left[x_{i}\right]}_{D}\right|}{\left|n_{B}^{\delta }(x_{i})\right|}). \end{array}$$
Since there exists $0\leq \frac{\left|\beta_{B}(D)\right|}{\left|U\right|}\leq \frac{1}{\left|U\right|}$, it follows that $n_{B}^{\delta }(x_{i})\cap {\left[x_{i}\right]}_{D}\subseteq n_{B}^{\delta }(x_{i})$ and $n_{B}^{\delta }(x_{i})\cap {\left[x_{i}\right]}_{D}\subseteq {\left[x_{i}\right]}_{D}$. Then, it is easily obtained that $\left|n_{B}^{\delta }(x_{i})\cap {\left[x_{i}\right]}_{D}\right|\leq \left|n_{B}^{\delta }(x_{i})\right|$ and $\left|n_{B}^{\delta }(x_{i})\cap {\left[x_{i}\right]}_{D}\right|\leq \left|{\left[x_{i}\right]}_{D}\right|$. Thus, obviously, $\frac{\left|n_{B}^{\delta }(x_{i})\cap {\left[x_{i}\right]}_{D}\right|}{\left|n_{B}^{\delta }(x_{i})\right|}\leq 1$ and $\frac{\left|n_{B}^{\delta }(x_{i})\cap {\left[x_{i}\right]}_{D}\right|}{\left|{\left[x_{i}\right]}_{D}\right|}\leq 1$ hold. Therefore, *H_δ_*(*D*, *B*) − *H_δ_*(*B*) ≥ 0 can be obtained, in essence, *H_δ_*(*D*, *B*) ≥ *H_δ_*(*B*). □

**Definition** **3.**
*Given a neighborhood decision system NS = (U, C, D,*
*δ*
*) with*
*B*
⊆
*C, and*
*any*
*a*
∈
*B, if H_δ_(D, B) ≤ H_δ_(D, B−{a}), then a is redundant in B with respect to D; otherwise, a is indispensable in B with respect to D. B is dependent if any attribute in B with respect to D is indispensable. B is called a reduct of C with respect to D if it satisfies the following two conditions:*
*(1)* 
*H_δ_(D, B) = H_δ_(D, C);*
*(2)* 
*H*
*_δ_*
*(D, B − {a}) < H*
*_δ_*
*(D, B), where*
*any a*
∈
*B.*



Obviously, a reduct of *C* with respect to *D* is the minimal attribute subset to retain the decision neighborhood entropy of *C* with respect to *D*.

**Definition** **4.**
*Given a neighborhood decision system NDS = (U, C, D, δ) with B*
⊆
*C and*
*any*
*attribute*
*a*
∈
*B, then the significance measure of a in B with respect to D is defined as*
(21)$$Sig_{in}(a,B,D)=H_{\delta }(D,B)-H_{\delta }(D,B-\{a\}).$$


**Definition** **5.**
*Given a neighborhood decision system NDS = (U, C, D, δ) with B*
⊆
*C and*
*any*
*attribute*
*a*
∈
*C − B, then the significance measure of a with respect to D is defined as*
(22)$$Sig_{out}(a,B,D)=H_{\delta }(D,B\cup \{a\})-H_{\delta }(D,B).$$


When *B* = ∅, *Sig**_out_*(*a*, *B*, *D*) = *H**_δ_*(*D*, {*a*}). From Definition 5, the significance of attribute *a* is the increment of the distinguishing information after adding *a* into *B*. The larger the value of *Sig*(*a*, *B*, *D*) is, the greater the importance of attribute *a* for *B* with respect to *D* is.

### 3.2. Attribute Reduction Algorithm Based on Decision Neighborhood Entropy

The process of the attribute reduction method for classification is shown in Figure 1.

To support efficient knowledge reduction, an attribute reduction algorithm based on decision neighborhood entropy (ARDNE) is constructed and described as Algorithm 1.
**Algorithm 1****Input:** A neighborhood decision system *NDS* = (*U*, *C*, *D*, *δ*)**Output:** A reduction attribute subset *R*1. Initialize *R* = ∅.2. while *Sig*(*S*, *R*, *D*) = 0 do3.     Let *Agent* = *R*, and *h* = 0.4.   for any *a*∈(*S* − *R*) do5.      Compute *H_δ_*(*D*, *R*∪{*a*}).6.      if *H_δ_*(*D*, *R*∪{*a*}) > *h* then7.       Let *Agent* = *R*∪{*a*}, and *h* = *H_δ_*(*D*, *B*∪{*a*}).8.      end if9.     end for10.    Let *R* = *Agent*.11. end while12. Let *r* = |*R*|.13. for *i* = 1 to *r* do14.   Select *a_i_*∈R.15.   Calculate *H_δ_*(*D*, *R* − {*a_i_*}).16.   if *H_δ_*(*D*, *R* − {*a_i_*}) ≥ *H_δ_*(*D*, *C*) then17.    Let *R = R* − {*a_i_*}.18. end19. Return a reduction attributes subset *R*.

### 3.3. Complexity Analysis of ARDNE Algorithm

From Algorithm 1, the decision neighborhood entropy and the neighborhood classes induced by the conditional attributes need to be frequently calculated in the computation of the attribute significance measure. The above computational process largely affects the time complexity of selecting attributes. Suppose that the number of attributes is *m*, and the number of samples is *n*. The complexity of calculating neighborhood classes is *O*(*mn*), and the computational complexity of calculating decision neighborhood entropy is *O*(*n*). Since *O*(*n*) < *O*(*mn*), the computational complexity of calculation of significance measure is *O*(*mn*). There are two loops in step 2 through step 11, then the worst time complexity of ARDNE is *O*(*m*^3^*n*). Suppose that the number of selected attributes is *m_R_*, for the calculation of the neighborhood classes, only the candidate attributes are considered instead of the entire attribute set. Then, the time complexity of achieving neighborhood classes is *O*(*m_R_n*). The outer loop times are *m_R_*, and the inner loop times are *m* − *m_R_*. Thus, the time complexity of this part is *O*(*m_R_*(*m* − *m**_R_*)*m_R_n*). Similar to the last steps, the time complexity of step 12 through step 18 is *O*(*m_R_**n*). It is well known that *m_R_* ≪ *m* in most cases. Therefore, the time complexity of ARDNE is close to *O*(*mn*). So far, ARDNE appears to be more efficient than some of the existing algorithms for attribute reduction in [33,39,40,41] in neighborhood decision systems. Furthermore, its space complexity is *O*(*mn*).

### 3.4. An Illustrative Example

In the following, the performance of the ARDNE algorithm is shown through an illustrative example in [42]. A neighborhood decision system *NDS* = (*U*, *C*, *D*, *δ*) is employed, where *U* = {*x*_1_, *x*_2_, *x*_3_, *x*_4_}, *C* = {*a*, *b*, *c*}, *D* = {*d*}, and *δ* = 0.3, as shown in Table 1.

For Table 1, an example for attribute reduction using Algorithm 1 is given. Then, the neighborhood class of each attribute in Table 1 is calculated by using the Euclidean distance function as follows. 

For an attribute subset {*a*}, one has that ∆_{*a*}_(*x*_1_, *x*_2_) = 0.09, ∆_{*a*}_(*x*_1_, *x*_3_) = 0.19, ∆_{*a*}_(*x*_1_, *x*_4_) = 0.49, ∆_{*a*}_(*x*_2_, *x*_3_) = 0.1, ∆_{*a*}_(*x*_2_, *x*_4_) = 0.4, and ∆_{*a*}_(*x*_3_, *x*_4_) = 0.3. Then, the neighborhood classes of any *x_i_*∈*U* can be computed by $n_{\left\{a\right\}}^{\delta }(x_{1})=\{x_{1},x_{2},x_{3}\}$, $n_{\left\{a\right\}}^{\delta }(x_{2})=\{x_{1},x_{2},x_{3}\}$, $n_{\left\{a\right\}}^{\delta }(x_{3})=\{x_{1},x_{2},x_{3},x_{4}\}$, and $n_{\left\{a\right\}}^{\delta }(x_{4})=\{x_{3},x_{4}\}$.

Due to *D* = {*d*} in Table 1, it follows that *U*/{*d*} = {*X*_1_, *X*_2_} = {{*x*_1_, *x*_2_}, {*x*_3_, *x*_4_}}. Then, from Equation (20), one has that $H_{\delta }(D,\;\{a\})=-\frac{1}{4\times 4}(\log(\frac{2^{2}}{4\times 2})+\log(\frac{2^{2}}{4\times 2})+\log(\frac{2^{2}}{4\times 2})+\log(\frac{2^{2}}{4\times 2}))=0.0753.$

Similarly, *H_δ_*(*D*, {*b*}) = 0.0753, *H_δ_*(*D*, {*c*}) = 0.1505, *H_δ_*(*D*, {*a, b*}) = 0.0753, *H_δ_*(*D*, {*a*, *c*}) = 0.1505, *H_δ_*(*D*, {*b*, *c*}) = 0.1505, and *H_δ_*(*D*, {*a*, *b*, *c*}) = 0.1505.

From the above calculated results, it can be observed that *H_δ_*(*D*, {*c*}) > *H_δ_*(*D*, {*a*}) = *H_δ_*(*D*, {*b*}). Since the decision neighborhood entropy of {*c*} and *D* is maximum, the attribute *c* should be added to the candidate attribute set (i.e., *R* = {*c*}). By computing, one has that *Sig*(*C*, *R*, D) = *H_δ_*(*D*, *C*) − *H_δ_*(*D*, {*c*}) = 0, which satisfies the termination criterion. Thus, a selected attribute subset {*c*} is achieved.

## 4. Experimental Results and Analysis

### 4.1. Experiment Preparation

It is known that the objective of an attribute reduction algorithm usually has two aspects: One is to select a small attribute subset and the other is to maintain high classification accuracy. To demonstrate the classification performances of our proposed attribute reduction algorithm described in Section 3.2 on several public data sets, the more comprehensive results of all contrasted algorithms should be achieved and analyzed. The selected four UCI (University of California at Irvine) Machine Learning Repository data sets with low-dimensional attributes include Ionosphere, Wisconsin Diagnostic Breast Cancer (Wdbc), Wisconsin Prognostic Breast Cancer (Wpbc), and Wine, which were downloaded from https://archive.ics.uci.edu/ml/datasets.html. The selected seven microarray gene expression data sets with high-dimensional attributes included Brain_Tumor1, Diffuse Large B Cell Lymphoma (DLBCL), Leukemia, Small Round Blue Cell Tumor (SRBCT), Colon, Lung, and Prostate, where the four gene expression data sets (Brain_Tumor1, DLBCL, Leukemia and SRBCT) can be downloaded at http://www.gems-system.org, the Colon gene expression data set can be downloaded at http://eps.upo.es/bigs/datasets.html, the Lung data set can be downloaded at http://bioinformatics.rutgers.ed/Static/Supple-mens/CompCancer/datasets, and the Prostate gene expression data set can be downloaded at http://www.gems-system.org. All of the data sets above are summarized in Table 2.

The experiments were performed on a personal computer running Windows 7 with an Intel(R) Core(TM) i5-3470 CPU operating at 3.20 GH, and 4 GB memory. All the simulation experiments were implemented in MATLAB R2014a, and the *k*-nearest neighbors (KNN) classifier and the support vector machine (SVM) classifier were selected to verify the classification accuracy in WEKA software, where the parameter *k* in KNN was set to 3 and the linear kernel functions were selected in SVM. All of the following experimental comparisons for classification on the selected attributes are implemented using a 10-fold cross-validation with all the test data sets, where every data set is first randomly divided into ten portions which are the same size subset of data each other, one data subset is used as the testing data set, the rest nine data subsets are used as the training data set, and each of the ten data subsets only is employed exactly once as the testing data set; secondly, the operation of the cross-validation is repeated ten times; finally, the average of ten test results is as the obtained classification accuracy.

### 4.2. Effect of Different Neighborhood Parameter Values

Since the value of neighborhood parameter decides the granularity of data manipulation, which affects both the cardinality of the data set and the classification accuracy of the attribute subset, in this subsection, our experiments concern the number of selected attributes and the classification accuracy with the different neighborhood parameter values. Following the experimental techniques designed by Chen et al. [33], the number of selected attributes and the classification accuracy of selected attribute subset for the different neighborhood parameter values is discussed to obtain a suitable neighborhood parameter value and a better attribute subset. The classification results of the data sets given in Table 2 were obtained by using the ARDNE algorithm with the different neighborhood parameters, shown in Figure 2, where the horizontal coordinates denotes the neighborhood parameters with *δ*
∈ [0.05, 1] at intervals of 0.05, and the left and right vertical axes represent the classification accuracy and the number of selected attributes, respectively.

Figure 2a–k show the number of selected attributes and the classification accuracy of eleven data sets with the different neighborhood parameter values. For the Ionosphere data set in Figure 2a, the classification accuracy reached its maximum when the parameter was 0.3. As the parameter value continued to increase, the number of selected attributes decreased, resulting in a rapid decrease in classification accuracy. For the Wdbc data set in Figure 2b, when the parameter took values in the interval [0.05, 0.15], there was little change in the classification accuracy, and the number of selected attributes was less when the parameter was 0.15. For the Wine data set in Figure 2c, the classification accuracy reached its maximum when the parameter was 0.15. Similarly, for the Wpbc data set in Figure 2d, the classification accuracy reached its maximum when the parameter was 0.2. For the Brain_Tumor1 data set in Figure 2e, the classification accuracy achieved the maximum when the parameter was set as 0.15. For the Colon data set in Figure 2f, as the parameter value continued to increase, the number of selected attributes increased first and in turn decreased, and then the classification accuracy reached its maximum when the parameter was 0.05. For the DLBCL data set in Figure 2g, when the parameter took the values in the interval [0.15, 0.3], there was a slight difference in the classification accuracy, and then the number of selected attributes was less when the parameter was 0.15. For the Leukemia, Lung, Prostate, and SRBCT data sets in Figure 2h–k, the classification accuracy reached their maximum when the parameters were 0.1, 0.3, 0.5, and 0.25, respectively. In addition, when the neighborhood parameter was about 0.5, the number of selected attributes would be close to zero. Therefore, the appropriate neighborhood parameters of eleven data sets should take values in the interval [0.05, 0.5].

### 4.3. Classification Results of ARDNE Algorithm under Different Neighborhood Parameter Values

In this part of our experiments, by using the above selected neighborhood parameters in Section 4.2, the classification results of the raw data and the reduced data using Algorithm 1 on the eleven gene expression data sets in Table 2 could be obtained. Then, the number of the attributes selected by the ARDNE algorithm and the corresponding classification accuracy with SVM and KNN based on 10-fold cross validation are shown in Table 3, respectively. The corresponding neighborhood parameter values are listed in the last column. 

From Table 3, it can be found that our proposed algorithm can greatly reduce the attributes for all the data sets without loss of classification accuracy, and most of the redundant attributes are reduced. In the four low-dimensional data sets, the classification accuracy of the SVM and KNN classifiers were higher. In the seven high-dimensional gene expression data sets, the classification accuracy of the KNN classifier was higher than that of the raw data, while there were some differences in the classification accuracy of SVM classifier. On the KNN classifier, the classification accuracy of all the data sets was higher than that of the raw data. On the SVM classifier, the classification accuracy of the Brain_Tumor1 data set was 3% less than that of the raw data, and the classification accuracy of the Colon and Leukemia data sets were slightly lower than that of the raw data. It shows that the reduced attribute set can maintain the classification accuracy of the raw data. However, on the SVM classifier, for the Prostate data set, the classification accuracy was 8.7% less than that of the raw data. The reason is that some attributes with important information during reduction are lost. What is more, for the average classification accuracy, our ARDNE algorithm obtained 91.04% and 91.86% on the SVM and KNN classifiers, respectively, which was higher by 10% than that of raw data sets. Therefore, the proposed ARDNE algorithm was efficient in dimension reduction of low-dimensional and high-dimensional data sets.

### 4.4. Classification Results of UCI Data Sets with Low-Dimensions

This portion of our experiments was to evaluate the performance of our proposed algorithm in terms of classification accuracy, and the classification performance of the ARDNE algorithm was compared with those of the other four related state-of-the-art attribute reduction algorithms on the four UCI data sets, selected from Table 2. The algorithms used in the comparison included: (1) The classical rough set algorithm (RS) [34], (2) the neighborhood rough set algorithm (NRS) [49], (3) the covering decision algorithm (CDA) [50], and (4) the max-decision neighborhood rough set algorithm (MDNRS) [28]. Table 4 gives the numbers of selected attributes in the reduced data with the four different algorithms. Table 5 and Table 6 show the comparison results of classification accuracy using the four different methods.

From Table 4, comparing the numbers of selected attributes, the NRS, CDA, MDNRS, and ARDNE algorithms were all superior to the RS algorithm, but the ARDNE algorithm was slightly inferior to NRS, CDA, and MDNRS algorithms. From Table 5 and Table 6, it is obvious that the classification accuracy of the proposed ARDNE algorithm outperformed that of the other algorithms on most of UCI data sets, except for the Wpbc data set. Furthermore, the average classification accuracy of the ARDNE was the highest and greatly improved on the SVM and KNN classifiers. For the Wpbc data set, the number of attributes selected by the ARDNE algorithm was six, which was not far from the MDNRS algorithm, and its classification accuracy was 0.8% lower than the MDNRS algorithm on KNN, while the accuracy was 8% higher than the MDNRS algorithm on SVM. Meanwhile, for the Wpbc data set, the number of attributes selected by the RS was seven, and its classification accuracy was 1% lower than the ARDNE algorithm on KNN, while the accuracy was 0.6% higher than the ARDNE algorithm on SVM. For the RS, NRS, and CDA algorithms, the classification accuracy of the Wdbc and Wine data sets were unstable. The classification accuracy of the Wine data sets only was 40.23% on the SVM classifier, and the average classification accuracy of the NRS model was the lowest. It can be obtained that the classification accuracy of the ARDNE algorithm on the SVM and KNN classifiers were relatively steady. Based on the results in Table 4, it can be seen that some important information attributes were lost in the process of reduction for the RS, NRS, and CDA algorithms, resulting in the decrease of classification accuracy of the reduced data sets with fewer attributes. The experimental results show that our attribute reduction algorithm could greatly remove the redundant attributes, and improve the classification accuracy for most of the data sets.

### 4.5. Classification Results of Microarray Data Sets with High-Dimensions

This subsection of our experiments continued testing the classification performance of the ARDNE algorithm, compared with those of the other three state-of-the-art entropy-based attribute reduction algorithms on the five microarray gene expression data sets with high-dimensional attributes, selected from Table 2. The algorithms used in the comparison included: (1) The mutual entropy-based attribute reduction algorithm (MEAR) [50], (2) the entropy gain-based attribute reduction algorithm (EGAR) [33], and (3) the average decision neighborhood entropy-based attribute reduction algorithm (ADNEAR) [42]. The objective of these further experiments was to show the classification power of the proposed approach to gene selection. Table 7 and Table 8 show the number of selected genes and the classification accuracy of the five high-dimensional gene expression data sets with the KNN and SVM classifiers, respectively.

From Table 7 and Table 8 the ARDNE algorithm obtained 93.8% and 91.9% average classification accuracy on the KNN and SVM classifiers, respectively. The classification accuracy of genes selected by the MEAR, EGAR, and ADNEAR algorithms were far lower than that with the ARDNE algorithm. For the MEAR algorithm, since the process of discretization generally results in loss of extensive useful gene information, the MEAR algorithm acquired the lower classification accuracy. For the number of selected genes, there was no significant difference among the EGAR, ADNEAR, and ARDNE algorithms. However, the classification accuracy of the ARDNE algorithm was superior to the EGAR and ADNEAR algorithms. It shows that the proposed ARDNE algorithm was able to find the most informative genes for classification. For the Colon data set, the classification accuracy of ARDNE was 80.8%, which was slightly less than that of the MEAR algorithm. So, it indicates that MDNRS algorithm was greatly affected by the data set, and the classification results were not as stable as the ARDNE algorithm. For the SRBCT data set, the classification accuracy of the ARDNE algorithm was obviously higher than those of other algorithms, and its number of selected genes only was six. The result of the further experiments shows that the proposed method had significant classification ability on the five microarray gene expression data sets.

### 4.6. Classification Results of Dimensionality Reduction Methods on Gene Expression Data Sets

To further verify the classification performance of our proposed method, the eight methods were employed to evaluate the number of selected genes and the classification accuracy on the four gene expression data sets selected from Table 2. The ARDNE algorithm was compared with the seven related state-of-the-art dimensionality reduction methods, which included: (1) The sequential forward selection algorithm (SFS) [51], (2) the sparse group lasso algorithm (SGL) [52], (3) the adaptive sparse group lasso based on conditional mutual information algorithm (ASGL-CMI) [53], (4) the Spearman’s rank correlation coefficient algorithm (SC^2^) [44], (5) the gene selection algorithm based on fisher linear discriminant and neighborhood rough set (FLD-NRS) [39], (6) the gene selection algorithm based on locally linear embedding and neighborhood rough set algorithm (LLE-NRS) [40], and (7) the RelieF algorithm [41] combined with the NRS algorithm [49] (RelieF+NRS). The SVM classifier in the WEKA tool was used to do some simulation experiments. The number of selected genes and the classification accuracy are shown in Table 9 and Table 10 respectively, where the symbol (–) denotes no results obtained for Leukemia using the SGL and ASGL-CMI algorithms.

According to the experimental results in terms of the number of selected genes and the classification accuracy in Table 9 and Table 10, the differences among the eight methods could be clearly identified. For the SGL and ASGL-CMI methods, the number of selected genes was obviously higher than that the other six algorithms, and then the classification accuracy of the SGL and ASGL-CMI methods was not ideal. For some methods, such as the SFS, SC^2^, FLD-NRS, and ARDNE algorithms, the average number of selected genes was less than 10. It follows that our proposed ARDNE algorithm selected fewer genes than the SFS, LLE-NRS, and RelieF+NRS algorithms, and it was roughly the same as SC^2^ and FLD-NRS. For the Colon data set, the classification accuracy of the ARDNE algorithm was 81% which was slightly lower than the SGL, ASGL-CMI, FLD-NRS, and LLE-NRS methods, but for the Leukemia, Lung, and Prostate data sets, the classification accuracy of the ARDNE algorithm were 96.7%, 98.7%, and 85.8%, respectively, which were higher than the other methods. For the SFS, SC^2^, LLE-NRS, and RelieF+NRS algorithms, their classification results were not as stable as the ARDNE algorithm. Thus, the classification effect of the algorithm for the four gene expression data sets would be slightly different, but the average classification ability of the ARDNE algorithm would not be affected. As for the average classification accuracy, the ARDNE algorithm obtained the highest accuracy. Therefore, our method was an efficient dimensionality reduction technique for high-dimensional, large-scale microarray data sets.

### 4.7. Statistical Analysis

The final part of our experiments was to further demonstrate the statistical significance of the results, and the Friedman test [53] and the Bonferroni–Dunn test [54] are employed in this paper. 

The Friedman statistic is described as follows
(23)$$\chi_{F}^{2}=\frac{12N}{k(k+1)}(\sum_{i=1}^{k}R_{i}^{2}-\frac{k{\left(k+1\right)}^{2}}{4}),$$
(24)$$F_{F}=\frac{(N-1)\chi_{F}^{2}}{N(k-1)-\chi_{F}^{2}},$$
where *k* is the number of algorithms, *N* is the number of data sets, and *R_i_* is the average ranking of algorithm *i* over all the data sets. And the critical distance [55] is denoted as
(25)$$CD_{\alpha }=q_{\alpha }\sqrt{\frac{k(k+1)}{6N}}.$$
where *q_α_* is the critical tabulated value for the test and *α* is the significant level of Bonferroni–Dunn test.

In the following, based on the classification performance of the five attribute reduction algorithms in Table 4, Table 11 and Table 12 show the rankings of the five algorithms under the KNN and SVM classifiers. The values of the different evaluation measures under the KNN and SVM classifiers are shown in Table 13. Table 14 and Table 15 show the rankings of the four algorithms in Table 7 under the KNN and SVM classifiers. The values of the different evaluation measures under the KNN and SVM classifiers are shown in Table 16. Similarly, Table 17 shows the rankings of the eight attribute reduction algorithms in Table 10 under the SVM classifier. The values of the different evaluation measures under the SVM classifiers are shown in Table 18. 

Table 11, Table 12, Table 13, Table 14, Table 15, Table 16, Table 17 and Table 18 show that the proposed ARDNE algorithm was statistically superior to the other algorithms in summary. It can be easily seen from Table 13, Table 16, and Table 18 that the values of *F_F_* were 13 and 23.78 under the KNN classifier, respectively, and those of *F_F_* were 2.33, 6.31, and 0.9 under the SVM classifier, respectively. When the significant level *α* = 0.1, the critical value of *F*(4,12) was 2.48, *F*(3,12) was 2.61, and *F*(7,7) was 2.78. The critical value *q*_0.1_= 2.241 can be found in [55], and it could be easily calculated from Equation (25) that the values of *CD* were 2.506, 1.83, and 1.093, respectively.

## 5. Conclusions

Attribute reduction is one of the important steps in data mining and classification learning. A number of measures for calculating the distinguishment ability of attribute subsets have been developed in recent years. Considering its effectiveness, neighborhood entropy is widely employed and discussed to evaluate attributes in neighborhood rough sets. In this paper, an attribute reduction method using neighborhood entropy measures in neighborhood rough sets is proposed. With the strong complementarity between the algebra definition of attribute importance and the definition of information view, some neighborhood entropy-based uncertainty measures in neighborhood decision systems are studied. Then, the significance measure is presented by combining the credibility degree with the coverage degree to analyze the classification ability of the selected attribute subset. On the basis of these theories, a heuristic attribute reduction algorithm is developed for the dimensionality reduction task to solve the practical problem. On the four UCI data sets with low-dimensional attributes and the seven microarray gene expression data sets with high-dimensional attributes, a series of experiments are carried out for verifying the effectiveness of the proposed method. The experimental results indicate that our algorithm is effective to remove the most redundant attributes without loss of classification accuracy. Comparing with the other related reduction algorithms, the reduction ability and the classification accuracy are more superior for knowledge reduction.

## Figures and Tables

**Figure 1 entropy-21-00155-f001:**
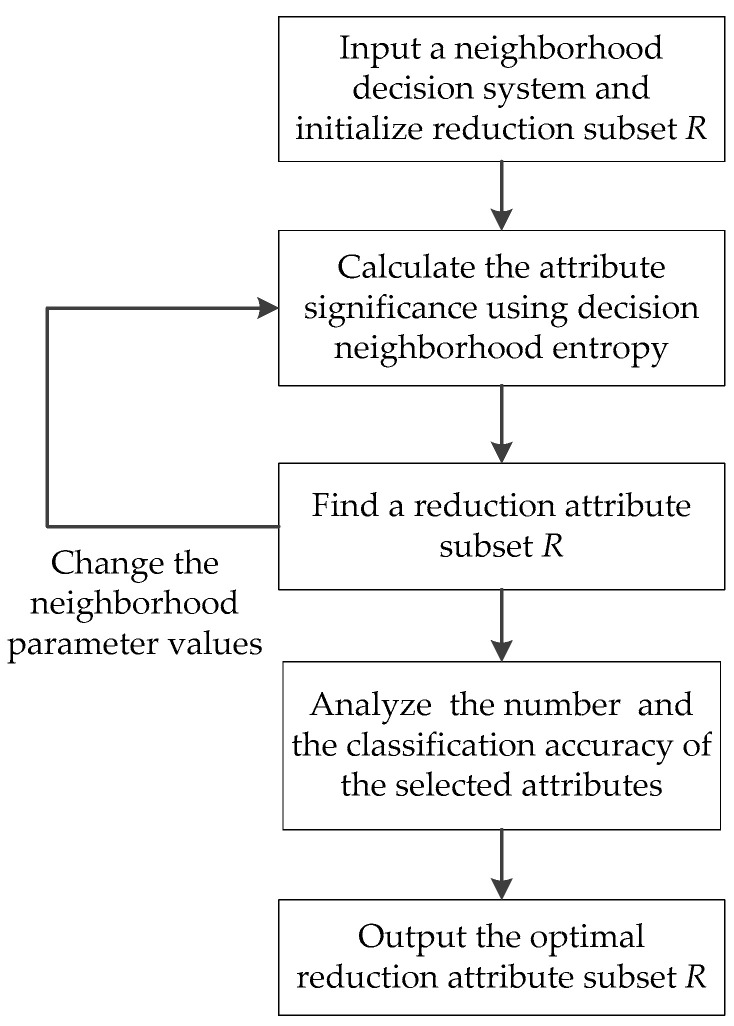
The process of the attribute reduction method based on decision neighborhood entropy.

**Figure 2 entropy-21-00155-f002:**
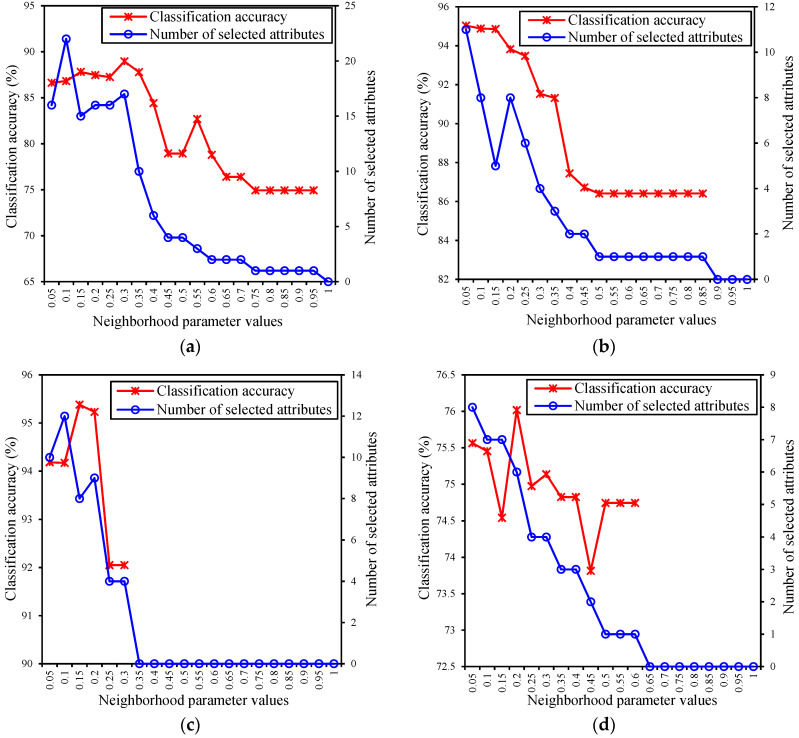
The number of selected attributes and the classification accuracy of the eleven data sets with the different neighborhood parameter values. (**a**) Ionosphere data set; (**b**) Wdbc data set; (**c**) Wine data set; (**d**) Wpbc data set; (**e**) Brain_Tumor1 data set; (**f**) Colon data set; (**g**) DLBCL data set; (**h**) Leukemia data set; (**i**) Lung data set; (**j**) Prostate data set; (**k**) SRBCT data set.

**Table 1 entropy-21-00155-t001:** A neighborhood decision system.

U	a	b	c	d
*x* _1_	0.12	0.41	0.61	Y
*x* _2_	0.21	0.15	0.14	Y
*x* _3_	0.31	0.11	0.26	N
*x* _4_	0.61	0.13	0.23	N

**Table 2 entropy-21-00155-t002:** Description of the eleven public data sets.

No.	Data Sets	Samples	Attributes	Classes	Reference
1	Ionosphere	351	33	2	Fen et al. [28]
2	Wdbc	569	31	2	
3	Wine	178	13	3	
4	Wpbc	198	33	2	
5	Brain_Tumor1	90	5920	5	Huang et al. [43]
6	Colon	62	2000	2	Mu et al. [44]
7	DLBCL	77	5469	2	Wang et al. [45]
8	Leukemia	72	7129	2	Dong et al. [46]
9	Lung	181	12,533	2	Sun et al. [47]
10	Prostate	136	12,600	2	
11	SRBCT	63	2308	4	Tibshirani et al. [48]

Wdbc—Wisconsin Diagnostic Breast Cancer; Wpbc—Wisconsin Prognostic Breast Cancer; DLBCL—Diffuse Large B Cell Lymphoma; and SRBCT—Small Round Blue Cell Tumor.

**Table 3 entropy-21-00155-t003:** The number of selected attributes and the classification accuracy under the SVM and KNN classifiers on the raw data and the reduced data with Algorithm 1.

Data Sets	Raw Data			Reduced Data using Algorithm 1			*δ*
	Attributes	SVM	KNN	Attributes	SVM	KNN	
Ionosphere	33	0.874	0.857	11	0.909	0.893	0.3
Wdbc	31	0.538	0.896	10	0.959	0.959	0.15
Wine	13	0.401	0.69	7	0.959	0.96	0.15
Wpbc	33	0.667	0.752	6	0.772	0.753	0.2
Brain_Tumor1	5920	0.86	0.783	13	0.83	0.897	0.15
DLBCL	2000	0.965	0.896	10	0.993	0.998	0.05
Colon	5469	0.811	0.776	5	0.808	0.818	0.15
Lung	7129	0.979	0.975	6	0.99	0.99	0.1
Leukemia	12,533	0.973	0.842	6	0.967	0.981	0.3
Prostate	12,600	0.916	0.796	3	0.829	0.858	0.5
SRBCT	2308	0.984	0.808	6	1	1	0.25
Average	4369.9	0.815	0.825	7.5	0.911	0.919	

SVM—support vector machine and KNN—*k*-nearest neighbors.

**Table 4 entropy-21-00155-t004:** The number of selected attributes of the five reduction algorithms on the four UCI data sets.

Data Sets	RS	NRS	CDA	MDNRS	ARDNE
Ionosphere	17	8	9	8	11
Wdbc	8	2	2	2	10
Wine	5	3	2	4	7
Wpbc	7	2	2	4	6
Average	9.25	3.75	3.75	4.5	8.5

RS—classical rough set algorithm; NRS—neighborhood rough set algorithm; CDA—covering decision algorithm; MDNRS—max-decision neighborhood rough set algorithm; and ARDNE— attribute reduction algorithm based on decision neighborhood entropy.

**Table 5 entropy-21-00155-t005:** Classification accuracy of the five reduction algorithms on the four UCI data sets with KNN.

Data Sets	RS	NRS	CDA	MDNRS	ARDNE
Ionosphere	0.866	0.859	0.848	0.891	0.893
Wdbc	0.911	0.923	0.923	0.930	0.959
Wine	0.863	0.752	0.727	0.911	0.960
Wpbc	0.743	0.738	0.738	0.761	0.753
Average	0.846	0.818	0.809	0.873	0.891

**Table 6 entropy-21-00155-t006:** Classification accuracy of the five reduction algorithms on the four UCI data sets with SVM.

Data Sets	RS	NRS	CDA	MDNRS	ARDNE
Ionosphere	0.881	0.872	0.878	0.870	0.909
Wdbc	0.589	0.595	0.595	0.861	0.959
Wine	0.640	0.402	0.643	0.910	0.959
Wpbc	0.778	0.757	0.757	0.692	0.772
Average	0.722	0.657	0.718	0.833	0.900

**Table 7 entropy-21-00155-t007:** Classification results of the four entropy-based reduction algorithms with KNN.

Data Sets	MEAR		EGAR		ADNEAR		ARDNE	
	Genes	Accuracy	Genes	Accuracy	Genes	Accuracy	Genes	Accuracy
Brain_Tumor1	2	0.683	8	0.667	9	0.711	13	0.897
Colon	5	0.77	5	0.540	5	0.555	5	0.817
DLBCL	2	0.765	20	0.752	7	0.757	10	0.998
Leukemia	3	0.928	3	0.587	3	0.587	6	0.981
SRBCT	4	0.537	8	0.503	8	0.503	6	1
Average	3.2	0.737	8.2	0.610	6.4	0.622	8	0.938

MEAR—mutual entropy-based attribute reduction algorithm; EGAR—entropy gain-based attribute reduction algorithm; and ADNEAR—average decision neighborhood entropy-based attribute reduction algorithm.

**Table 8 entropy-21-00155-t008:** Classification results of the four entropy-based reduction algorithms with SVM

Data Sets	MEAR		EGAR		ADNEAR		ARDNE	
	Genes	Accuracy	Genes	Accuracy	Genes	Accuracy	Genes	Accuracy
Brain_Tumor1	2	0.691	8	0.666	9	0.666	13	0.830
Colon	5	0.849	5	0.643	5	0.643	5	0.808
DLBCL	2	0.777	20	0.862	7	0.862	10	0.993
Leukemia	3	0.920	3	0.536	3	0.536	6	0.967
SRBCT	4	0.539	8	0.535	8	0.535	6	1
Average	3.2	0.755	8.2	0.648	6.4	0.648	8	0.919

**Table 9 entropy-21-00155-t009:** The number of selected genes of the eight reduction algorithms on the four gene expression data sets.

Data Sets	SFS	SGL	ASGL-CMI	SC^2^	FLD-NRS	LLE-NRS	RelieF+NRS	ARDNE
Colon	19	55	33	4	6	16	9	5
Leukemia	7	-	-	5	6	22	17	6
Lung	3	43	32	3	3	16	23	6
Prostate	3	34	29	5	4	19	16	3
Average	8	44	31.3	4.25	4.75	18.25	16.25	5

SFS—sequential forward selection algorithm; SGL—sparse group lasso algorithm; ASGL-CMI— adaptive sparse group lasso based on conditional mutual information algorithm; SC^2^—Spearman’s rank correlation coefficient algorithm; FLD-NRS—gene selection algorithm based on fisher linear discriminant and neighborhood rough set; LLE-NRS—gene selection algorithm based on locally linear embedding and neighborhood rough set algorithm, and RelieF+NRS—RelieF algorithm combined with NRS algorithm.

**Table 10 entropy-21-00155-t010:** The classification accuracy of the eight reduction algorithms on the four gene expression data sets.

Data Sets	SFS	SGL	ASGL-CMI	SC^2^	FLD-NRS	LLE-NRS	RelieF+NRS	ARDNE
Colon	0.521	0.826	0.851	0.805	0.88	0.84	0.564	0.81
Leukemia	0.969	-	-	0.852	0.828	0.868	0.563	0.967
Lung	0.833	0.827	0.841	0.806	0.889	0.907	0.919	0.987
Prostate	0.840	0.834	0.858	0.795	0.8	0.711	0.642	0.858
Average	0.791	0.829	0.85	0.815	0.849	0.832	0.672	0.898

**Table 11 entropy-21-00155-t011:** Ranking of the five attribute reduction algorithms with KNN.

Data Sets	RS	NRS	CDA	MDNRS	ARDNE
Ionosphere	3	4	5	2	1
Wdbc	5	3.5	3.5	2	1
Wine	3	4	5	2	1
Wpbc	3	4.5	4.5	1	2
Average	3.5	4	4.5	1.75	1.25

**Table 12 entropy-21-00155-t012:** Ranking of the five attribute reduction algorithms with SVM.

Data Sets	RS	NRS	CDA	MDNRS	ARDNE
Ionosphere	2	4	3	5	1
Wdbc	5	3.5	3.5	2	1
Wine	4	5	3	2	1
Wpbc	1	3.5	3.5	5	2
Average	3	4	3.25	3.5	1.25

**Table 13 entropy-21-00155-t013:** *F_F_* Values for the two classifiers.

	KNN	SVM
$\chi_{F}^{2}$	13	7
*F_F_*	13	2.33

*F_F_*—Iman-Davenport test and $\chi_{F}^{2}$—Friedman statistics.

**Table 14 entropy-21-00155-t014:** Ranking of the four attribute reduction algorithms with KNN.

Data Sets	MEAR	EGAR	ADNEAR	ARDNE
Brain_Tumor1	3	4	2	1
Colon	2	4	3	1
DLBCL	2	4	3	1
Leukemia	2	3.5	3.5	1
SRBCT	2	3.5	3.5	1
Average	2.2	3.8	3	1

**Table 15 entropy-21-00155-t015:** Ranking of the four attribute reduction algorithms with SVM.

Data Sets	MEAR	EGAR	ADNEAR	ARDNE
Brain_Tumor1	2	3.5	3.5	1
Colon	1	3.5	3.5	2
DLBCL	4	2.5	2.5	1
Leukemia	2	3.5	3.5	1
SRBCT	2	3.5	3.5	1
Average	2.2	3.3	3.3	1.2

**Table 16 entropy-21-00155-t016:** *F_F_* Values for the two classifiers.

	KNN	SVM
$\chi_{F}^{2}$	12.84	9.18
*F_F_*	23.78	6.31

**Table 17 entropy-21-00155-t017:** Ranking of the eight attribute reduction algorithms with SVM.

Data Sets	SFS	SGL	ASGL-CMI	SC^2^	FLD-NRS	LLE-NRS	RelieF+NRS	ARDNE
Lung	6	7	5	8	4	3	2	1
Prostate	3	4	1.5	6	5	7	8	1.5
Average	4.5	5.5	3.25	7	4.5	5	5	1.25

**Table 18 entropy-21-00155-t018:** *F_F_* Values for the two classifiers.

	SVM
$\chi_{F}^{2}$	6.63
*F_F_*	0.9

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
