# Peer review of "An Attribute Reduction Method Using Neighborhood Entropy Measures in Neighborhood Rough Sets"

_entropy, 2019, doi:10.3390/e21020155_

Round 1

Reviewer 1 Report

The article is interesting and provide good findings though it is not very novel but the experiments provides good coverage on the potential of the work in attribute reduction.

The article need to have a better highlight on the novelty in terms of better coverage and comparison with recent work.  The paper shoudl revise its references as there are many citation some are outdated and other not relevant. 

You may look to other issues in terms of Neighborhood Covering

"Fuzzy Neighborhood Covering for Three-Way Classification", Information Sciences, 2018 https://www.sciencedirect.com/science/article/pii/S0020025518305929

"An incremental attribute reduction method for dynamic data mining”, Information Sciences, Volume 465, October 2018, Pages 202-218, https://www.sciencedirect.com/science/article/pii/S002002551830519X

"Incremental rough set approach for hierarchical multicriteria classification", Information Sciences, Volume 429, March 2018, Pages 72–87. https://www.sciencedirect.com/science/article/pii/S0020025517310691

Author Response

Dear,

We are very grateful to you for your valuable comments and suggestions. We have carefully revised the paper in accordance with these comments and suggestions. The added and modified parts are shown in red in the revised manuscript (and changes are marked). The main revisions are as follows.

Comment: The article need to have a better highlight on the novelty in terms of better coverage and comparison with recent work. The paper should revise its references as there are many citation some are outdated and other not relevant.

Responses: Thank you very much for your valuable suggestion.

First, for better coverage and comparison with recent work, we have presented some comparative experiments with the literatures published from 2016 to 2018 in the experiments. These literatures are listed as follows:

38. Sun, L.; Zhang, X.Y.; Xu, J.C.; Wang, W.; Liu, R.N. A gene selection approach based on the fisher linear discriminant and the neighborhood rough set. Bioengineered 2018, vol. 9, no. 1, pp. 144−151.

39. Sun, L.; Xu, J.C.; Wang, W.; Yin, Y. Locally linear embedding and neighborhood rough set-based gene selection for gene expression data classification. Genetics and Molecular Research 2016, vol. 15, no. 3, article ID: UNSP 15038990.

40. Urbanowicz, R.J.; Meeker, M.; La Cava, W.; Olsona, R.S.; Moore, J.H. Relief-based feature selection: introduction and review. Journal of Biomedical Informatics 2018, vol. 85, pp. 189−203.

41. Sun, L.; Zhang, X.Y.; Qian, Y.H.; Xu, J.C.; Zhang, S.G.; Tian, Y. Joint neighborhood entropy-based gene selection method with fisher score for tumor classification. Applied Intelligence 2018, doi: 10.1007/s10489-018-1320-1.

43. Xu, J.C.; Mu, H.Y.; Wang, Y.; Huang, F.Z. Feature genes selection using supervised locally linear embedding and correlation coefficient for microarray classification. Computational and Mathematical Methods in Medicine 2018, vol. 2018, article ID: 5490513.

52. Li, J.T.; Dong, W.P.; Meng, D.Y. Grouped gene selection of cancer via adaptive sparse group lasso based on conditional mutual information. IEEE/ACM Transactions on Computational Biology & Bioinformatics 2017, doi:10.1109/tcbb.2017.2761871.

Second, we have revised the references, and deleted many outdated and irrelevant citations as follows:

1. Hu, Q.H.; Liu, J.F.; Yu, D.R. Mixed feature selection based on granulation and approximation. Knowledge-Based Systems 2008, vol. 21, no. 4, pp. 294−304.

2. Hasanloei, M.A.V.; Sheikhpour, R.; Sarram, M.A.; Sheikhpour, E.; Sharifi, H. A combined Fisher and Laplacian score for feature selection in QSAR based drug design using compounds with known and unknown activities. Journal of Computer-Aided Molecular Design 2018, vol. 32, no. 2, pp. 375−384.

3. Lyu, H.Q.; Wan, M.X.; Han, J.Q.; Liu, R.L.; Wang, C. A filter feature selection method based on the Maximal Information Coefficient and Gram-Schmidt Orthogonalization for biomedical data mining. Computers in Biology and Medicine 2017, vol. 89, pp. 264−274.

4. Xue, B.; Zhang, M.J.; Browne, W.N.; Yao, X. A survey on evolutionary computation approaches to feature selection. IEEE Transactions on Evolutionary Computation 2016, vol. 20, no. 4, pp. 606−626.

5. Swiniarski R.W.; Skowron A. Rough set methods in feature selection and recognition. Pattern Recognition Letters 2003, vol. 24, no. 6, pp. 833−849.

6. Zhang, J.B.; Li, T.R.; Ruan, D.; Liu, D. Neighborhood rough sets for dynamic data mining. International Journal of Intelligent Systems 2012, vol. 27, no. 4, pp. 317−342.

7. Marinaki, M.; Marinakis, Y. A bumble bees mating optimization algorithm for the feature selection problem. International Journal of Machine Learning and Cybernetics 2016, vol. 7, no. 4, pp. 519−538.

8. Jia, X.Y.; Liao, W.H.; Tang, Z.M.; Shang, L. Minimum cost attribute reduction in decision-theoretic rough set model. Information Sciences 2013, vol. 219, pp. 151−167.

9. Zhao, H.; Min, F.; Zhu, W. A backtracking approach to minimal cost feature selection of numerical data, Journal of Chemical Information and Computer Sciences 2013, vol. 10, no. 13, pp. 4105−4115.

10. Jiang, S.Y.; Lu, Y.S. Two new reduction definitions of decision table. Journal of Chinese Computer Systems 2006, vol. 27, no. 3, pp. 512−515.

11. Liang, J.Y.; Shi, Z.Z.; Li, D.Y.; Wierman, M.J. Information entropy, rough entropy and knowledge granulation in incomplete information systems. International Journal of general systems 2006, vol. 35, no. 6, pp. 641−654.

12. Hu, Q.H.; Yu, D.R.; Xie, Z.X. Numerical attribute reduction based on neighborhood granulation and rough approximation. Chinese Journal of Software 2008, vol. 19, no. 3, pp. 640−649.

13. Ye, M.Q.; Wu, X.D.; Hu, X.G.; Hu, D.H. Multi-level rough set reduction for decision rule mining. Applied intelligence 2013, vol. 39, no. 3, pp. 642−658.

14. Zhang, B.W.; Min, F.; Ciucci, D. Representative-based classification through covering-based neighborhood rough sets. Applied Intelligence 2015, vol. 43, no. 4, pp. 840−854.

15. Yao, Y.Y.; Yao, B.X. Covering based rough set approximations. Information Sciences 2012, vol. 200, pp. 91–107.

16. Shakiba A.; Hooshmandasl M.R. Data volume reduction in covering approximation spaces with respect to twenty-two types of covering based rough sets. International Journal of Approximate Reasoning 2016, vol. 75, pp. 13−38.

17. Chen, Y.M.; Zeng, Z.Q.; Lu, J.W. Neighborhood rough set reduction with fish swarm algorithm. Soft Computing 2017, vol. 21, no. 23, pp. 6907−6918.

18. Liu, Y.; Huang, W.L.; Jiang, Y.L.; Zeng, Z.Y. Quick attribute reduct algorithm for neighborhood rough set model. Information Sciences 2014, vol. 271, pp. 65–81.

19. Yang, M.; Chen, Y.J.; Ji, G.L. Semi Fisher Score: A semi-supervised method for feature selection. Proceeding of International Conference on Machine Learning and Cybernetics 2010, vol. 1, pp. 527−532.

Third, we have added the assigned relevant references as follows:

5. Jing, Y.G.; Li, T.R.; Fujita, H.; Wang, B.L.; Cheng, N. An incremental attribute reduction method for dynamic data mining. Information Sciences 2018, vol. 465, pp. 202−218.

20. Luo, C.; Li, T.R.; Chen, H.M.; Fujita, H.; Yi, Z. Incremental rough set approach for hierarchical multicriteria classification. Information Sciences, 2018, vol. 429, pp. 72−87.

25. Yue, X.D.; Chen, Y.F.; Miao, D.Q.; Fujita, H. Fuzzy neighborhood covering for three-way classification. Information Sciences 2018, doi: 10.1016/j.ins.2018.07.065.

Thank you once again for your constructive and valuable comments.

Best wishes,

Prof. Lin Sun, Ph.D.,

College of Computer and Information Engineering, Henan Normal University

Email: linsunok@gmail.com

Reviewer 2 Report

This paper proposed a complementary method based on classical Rough Set Theory and Entropy for feature selection named Attribute Reduction based on Decision Neighborhood  Entropy  (ARDNE). The experiments have been well conducted; nevertheless, I consider that some adjustments could be carried out.

1. The hypothesis in the introduction could be improved.

2. In line 94, 95, 96: the sentence "The conditional attributes with higher credibility... " is repeated in line104-105.

3. Check equation (2).

4. The illustrative example of section 3.4 no is clear completely. I consider that the author can explain or correct the expression of line 339. The sum has all elements repeated. I verified the expression using the data shown in Table 1 and the results of line 333 and 334, however, these did not coincide.

5. Finally, I suggest that the authors could use statistical measures for comparing the different algorithm respect the proposed methods. Besides, they can explain, why in all experiments were not applied the same algorithms for the comparison.

Author Response

Dear,

We are very grateful to you for your valuable comments and suggestions. We have carefully revised the paper in accordance with these comments and suggestions. The added and modified parts are shown in red in the revised manuscript (and changes are marked). The main revisions are as follows.

Comment 1: The hypothesis in the introduction could be improved.

Responses: Thank you very much for your valuable suggestion.

We have revised the introduction of this paper carefully.

Comment 2: In line 94, 95, 96: the sentence "The conditional attributes with higher credibility... " is repeated in line104-105.

Responses: Thank you very much for your valuable suggestion.

On Page 3: The repeated sentence “The conditional attributes with higher credibility and coverage degrees are more important with respect to the decision attribute.” has been deleted.

Comment 3: Check equation (2).

Responses: Thank you very much for your valuable suggestion.

We have checked Eq. (2) carefully.

Comment 4: The illustrative example of section 3.4 no is clear completely. I consider that the author can explain or correct the expression of line 339. The sum has all elements repeated. I verified the expression using the data shown in Table 1 and the results of line 333 and 334, however, these did not coincide.

Responses: Thank you very much for your valuable suggestion.

We have checked and revised the illustrative example of Subsection 3.4, shown as follows:

On Pages 10 and 11: A neighborhood decision system NDS = (U, C, D, δ) is employed, where U = {x1, x2, x3, x4}, C = {a, b, c}, D = {d}, and δ = 0.3, as shown in Table 1.

Table 1. A neighborhood decision system

U

a

b

c

d

x1

0.12

0.41

0.61

Y

x2

0.21

0.15

0.14

Y

x3

0.31

0.11

0.26

N

x4

0.61

0.13

0.23

N

For Table 1, an example of attribute reduction using Algorithm 1 is given. Then, the neighborhood class of each attribute in Table 1 is calculated by using the Euclidean distance function as follows.

For an attribute subset {a}, one has that ∆{a}(x1, x2) = 0.09, ∆{a}(x1, x3) = 0.19, ∆{a}(x1, x4) = 0.49, ∆{a}(x2, x3) = 0.1, ∆{a}(x2, x4) = 0.4, and ∆{a}(x3, x4) = 0.3. Then, the neighborhood classes of any xiU can be computed by

, , , and .

Due to D = {d} in Table 1, it follows that U/{d} = {X1, X2} = {{x1, x2}, {x3, x4}}. Then, one has that

Similarly, Hδ(D, {b}) = 0.0753, Hδ(D, {c}) = 0.1505, Hδ(D, {a, b}) = 0.0753, Hδ(D, {a, c}) = 0.1505, Hδ(D, {b, c}) = 0.1505, and Hδ(D, {a, b, c}) = 0.1505.

From the above calculated results, it can be observed that Hδ(D, {c}) > Hδ(D, {a}) = Hδ(D, {b}). Since the decision neighborhood entropy of {c} and D is maximum, the attribute c should be added to the candidate attribute set, i.e., R = {c}. By computing, we have that Sig(C, R, D) = Hδ(D, C) − Hδ(D, {c}) = 0.1505 − 0.1505 = 0, which satisfies the termination criterion. Thus, a selected attribute subset {c} is achieved.

Comment 5: Finally, I suggest that the authors could use statistical measures for comparing the different algorithm respect the proposed methods. Besides, they can explain, why in all experiments were not applied the same algorithms for the comparison.

Responses: Thank you very much for your valuable suggestion.

First, on Page3 20-22, we have added the subsection 4.7 to compare different algorithms with the proposed method by statistical measures.

4.7 Statistical analysis

The final part of our experiments is to further demonstrate the statistical significance of the results, and the Friedman test [53] and the Bonferroni-Dunn test [54] are employed in this paper. The Friedman statistic is described as follows

,

(23)

,

(24)

where k is the number of algorithms, N is the number of data sets, and Ri is the average ranking of algorithm i over all the data sets. FF follows a Fisher distribution with k−1 and (k−1)(N−1) degrees of freedom. Based on above test results, if the average level of distance exceeds the critical distance, the performance of the two algorithms will be significantly different. The critical distance [55] is denoted as

.

(25)

where qα is the critical tabulated value for the test and α is the significant level of Bonferroni-Dunn test.

In the following, the Friedman tests are introduced to investigate whether the classification performance of the five attribute reduction algorithms (RS, NRS, CDA, MDNRS and ARDNE) in Table 4 are significantly different. Tables 11 and 12 show the rankings of the five attribute reduction algorithms under the KNN and SVM classifiers. The values of the different evaluation measures under the KNN and SVM classifiers are shown in Table 13.

To illustrate the statistical differences of the four attribute reduction algorithms (MEAR, EGAR, ADNEAR and ARDNE) in Table 7, Tables 14 and 15 show the rankings of the four algorithms under the KNN and SVM classifiers. The values of the different evaluation measures under the KNN and SVM classifiers are shown in Table 16. Similarly, Table 17 shows the rankings of the eight attribute reduction algorithms (SFS, SGL, ASGL-CMI, SC2, FLD-NRS, LLE-NRS, RelieF + NRS and ARDNE) in Table 10 under the SVM classifier. The values of the different evaluation measures under the SVM classifiers are shown in Table 18.

Table 11. Ranking of the five attribute reduction algorithms with KNN

Data sets

RS

NRS

CDA

MDNRS

ARDNE

Ionosphere

3

4

5

2

1

Wdbc

5

3.5

3.5

2

1

Wine

3

4

5

2

1

Wpbc

3

4.5

4.5

1

2

Average

3.5

4

4.5

1.75

1.25

Table 12. Ranking of the five attribute reduction algorithms with SVM

Data sets

RS

NRS

CDA

MDNRS

ARDNE

Ionosphere

2

4

3

5

1

Wdbc

5

3.5

3.5

2

1

Wine

4

5

3

2

1

Wpbc

1

3.5

3.5

5

2

Average

3

4

3.25

3.5

1.25

Table 13. FF Values for the two classifiers

KNN

SVM

13

7

FF

13

2.33

Table 14. Ranking of the four attribute reduction algorithms with KNN

Data sets

MEAR

EGAR

ADNEAR

ARDNE

Brain_Tumor1

3

4

2

1

Colon

2

4

3

1

DLBCL

2

4

3

1

Leukemia

2

3.5

3.5

1

SRBCT

2

3.5

3.5

1

Average

2.2

3.8

3

1

Table 15. Ranking of the four attribute reduction algorithms with SVM

Data sets

MEAR

EGAR

ADNEAR

ARDNE

Brain_Tumor1

2

3.5

3.5

1

Colon

1

3.5

3.5

2

DLBCL

4

2.5

2.5

1

Leukemia

2

3.5

3.5

1

SRBCT

2

3.5

3.5

1

Average

2.2

3.3

3.3

1.2

Table 16. FF Values for the two classifiers

KNN

SVM

12.84

9.18

FF

23.78

6.31

Table 17. Ranking of the eight attribute reduction algorithms with SVM

Data sets

SFS

SGL

ASGL-CMI

SC2

FLD-NRS

LLE-NRS

RelieF+NRS

ARDNE

Lung

6

7

5

8

4

3

2

1

Prostate

3

4

1.5

6

5

7

8

1.5

Average

4.5

5.5

3.25

7

4.5

5

5

1.25

Table 18. FF Values for the two classifiers

SVM

6.63

FF

0.9

Tables 11-18 show that the proposed ARDNE algorithm is statistically superior to the other algorithms on the whole. It can be easily seen from Tables 13, 16 and 18 that the values of FF are 13 and 23.78 under the KNN classifier, respectively, and the values of FF are 2.33, 6.31 and 0.9 under the SVM classifier, respectively. When the significant level α = 0.1, the critical value of F(4,12) is 2.48, F(3,12) is 2.61, and F(7,7) is 2.78. Since the Friedman test rejects null hypothesis under the two classifiers when all algorithms have the same performance. Therefore, the two Bonferroni-Dunn tests need to be performed. The critical value q0.1= 2.241 can be find in [55], and it can be easily calculated from Eq. (25) that the values of CD are 2.506, 1.83 and 1.093, respectively.

Second, following the experimental techniques designed in [28, 34, 48, 49], the same data sets (Ionosphere, Wdbc, Wine and Wpbc) in Table 1 can be used for comparing the classification performance of the ARDNE algorithm with these algorithms. Similarly, following the experimental techniques in [33, 41, 49], the same data sets (Brain_Tumor1, Colon, DLBCL, Leukemia and SRBCT) can be found, so we applied these algorithms (MEAR, EGAR, ADNEAR and ARDNE) on the same data sets for the comparison. In the subsection 4.6, some experimental techniques in [38-40, 43, 48, 50-52] are applied on the four same data sets (Colon, Leukemia, Lung and Prostate) to obtain the comparison results. Thus, the compared algorithms in Subsection 4.2-4.7 were slightly different.

Thank you once again for your constructive and valuable comments.

Best wishes,

Prof. Lin Sun, Ph.D.,

College of Computer and Information Engineering, Henan Normal University

Email: linsunok@gmail.com

Round 2

Reviewer 2 Report

All corrections were attended.